# Towards Earth System Modeling: Coupled Ocean Forecasting

Ségolène Berthou[1], John Siddorn[2], Vivian Fraser-Leonhardt[1], Pierre-Yves Le Traon[3], Ibrahim Hoteit[4]

[1]MetOffice, Exeter, UK
[2]Data, Science and Technology, National Oceanography Centre, Southampton, UK
[3]Mercator Ocean International, Toulouse, France
[4]Physical Science and Engineering Division, King Abdullah University of Science and Technology (KAUST), Thuwal, Saudi Arabia

*Correspondence to*: Ségolène Berthou (segolene.berthou@metoffice.gov.uk)

**Abstract.** Forecasting across different earth system components has initially been achieved independently, but increasing computer power, increasing model accuracy, increasing connectivity between experts and increasing need for multi-hazard weather warning is changing the scene. Coupling methods, which involve exchanging information between discrete modelling systems, enable to gain accuracy and consistency across earth system components. The paper explains the principles of two-way coupling, where models run simultaneously and exchange information both ways. As individual models reach better accuracy, coupling becomes a key factor to improve forecasting capability because it reproduces the natural complexity of the environment: a wealth of literature shows the benefits of coupling. However, coupling is still limited in operational oceanography by its large demands on computational resources, by data assimilation techniques (currently not very well harmonized between the different models) and by administrative separation of forecasts across different earth system components. Overcoming these barriers will support ocean predictions towards a multi-hazard approach and a more accurate representation of the earth systems component interactions and improve collaborations between multi-disciplinary forecasting communities.

## 1 Introduction

Coupling can be loosely defined as the process of exchanging information between discrete modelling systems, generally of components of the earth system, to better represent exchange processes (Shapiro et al., 2010). The number of components of a coupled system, and indeed the level of coupling between the components, varies depending on the application. Coupled global climate models (GCMs) generally include the ocean, ice, atmosphere and land surface. Increasingly surface waves are included to represent the exchange between the ocean and the atmosphere better, especially for applications that require representation of natural hazards such as storms. For earth system models which need to include predictions of the biogenic components to predict carbon and other nutrient transfers the components are often extended to include ocean biogeochemistry and atmospheric chemistry (Mulcahy et al., 2023).

There are a number of solutions to how this coupling may be achieved, and which is preferred will depend both on the scientific importance of the exchanges and the timescales on which they occur and on technical limitations. In the "traditional" way of working the models are run independently with a flux of information from adjacent components of the earth system being calculated based on independent and non-interactive models. This implies that the winds, precipitation and air temperatures ("forcing") used to drive the exchanges at the ocean's surface do not respond to changes in the ocean conditions themselves. The forcing is not calculated on a timestep basis but over a period generally somewhere between an hour and a day. Forecasts run in this mode are termed forced or one-way coupled.

Coupled systems exist with varying complexity of exchanges between models. For example, a common approach for the coupling of hydrodynamics and sea ice is to run both systems at the same time and exchange information both ways. These are termed fully or two-way coupled systems. In these two-way coupled systems, the independent models often communicate with each other through an interface code ("coupler") which allows the independent models to operate on different grids and with different timesteps (Larson et al. 2005, Valcke, 2013, Hanke et al. 2016). As the number of components interacting with each other increases the flexibility of including a coupler becomes increasingly attractive. A coupling software creates a computational interface between separate systems that allows the passing of information between them without undue intrusion into the code of the modelling systems. This approach is widely used (e.g. Lewis et al., 2018; Pianezze et al., 2022; Wahle et al., 2017) but other approaches exist. ECMWF (Wedi et al., 2015) have integrated their various modelling components into a single executable with the passing of information being done internally within the code rather than through a separate coupling software. Figure 1 illustrates the Regional Environmental Prediction system under development in the United Kingdom, with complex exchanges between five different models, using three different coupling approaches (Best et al., 2004, Valcke 2005, Bruggeman and Bolding, 2014).

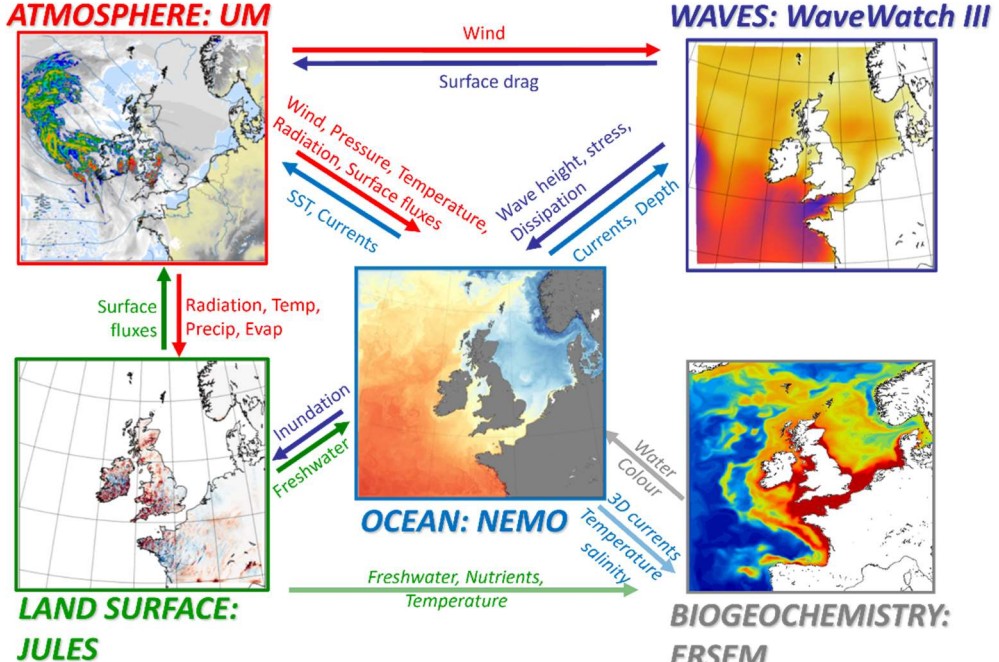

**55**

**Figure 1: Regional coupled system under development in the United Kingdom for the Regional Environmental Prediction project (Lewis et al., 2019), bringing together all the models run by the Met Office for short-term predictions and climate projections. Arrows represent exchanges between models – either as integrated coupling at the time-step (Best et al. 2004) (UM/JULES), 2D coupling through the OASIS coupler (Valckle et al. 2013) (UM / WaveWatch III / NEMO) or 3D coupling through the FABM**
**60** **coupler (Bruggeman and Bolding, 2014) (NEMO/ERSEM).**

## 2 Why is coupling important for Ocean Prediction?

Atmosphere / ocean coupling is common practice at the seasonal and decadal timescales. At these scales, most of the memory is contained in the ocean and in coupled interactions, such as for the El Nino Southern Oscillation (ENSO). Indeed, both the ocean and atmosphere can propagate an anomaly in the other component to remote places. For example, oceanic equatorial
**65** waves generated by wind anomalies can propagate to the whole tropical Pacific and generate an El Nino event, and in turn the atmosphere may generate teleconnections from the tropics to the mid-latitudes through upper-level Rossby wave trains in the troposphere or planetary waves in the stratosphere, and influence the ocean back in remote ocean basins (Hardiman et al., 2019; Kim et al., 2012). These may take longer than 10-days to propagate and are therefore sources of seasonal and multi-annual forecast signals. For short-term marine prediction, coupling is emerging as a new potential for improving both
**70** atmospheric and oceanic predictions (Brassington et al., 2015).

A clear and extremely well documented weather situation when air-sea coupling is key for both atmosphere and ocean are tropical cyclone forecasts: the strength of tropical cyclones is decreased through large decreases in sea surface temperature (SST) caused by intense turbulent fluxes, deepening of the surface mixed layer by entrainment (Vellinga et al., 2020; Mogensen

et al., 2017; Castillo et al., 2022; Feng et al., 2019) and (if the cyclone translation speed is slow) by upwelling (Corale et al., 2023; Yablonsky et al., 2009). In more general situations, coupling reduces the lifetime of mesoscale eddies and dampens submesoscale currents through dampening of the wind stress curl and heat fluxes (Yang et al.,2019; Renault et al., 2016; Renault et al., 2018; Dawe and Thompson, 2006). Coupling also sometimes involves a higher resolution atmosphere than forcing, which then results in more turbulent eddy kinetic energy in the ocean (Storto et al., 2023). In the tropics, dynamical waves in the atmosphere and ocean can influence each other. For example, Madden Julian Oscillation (MJO) atmospheric events in the Indian Ocean can be modulated by coupling (Fu et al., 2017) or simply by the diurnal cycle of SST (Karlowska et al., 2023). Convectively Coupled Kelvin waves also generate a strong signal in the Indian ocean (Azaneu et al., 2021).

At the coastal scale, coupling also becomes interesting since the assumptions of equilibrium between earth system components often break down (e.g. wave state is not in equilibrium with winds in the sheltered North Sea - Grayek et al., 2023; Wiese et al., 2019; Whale et al., 2017). Some examples in the literature include better near-surface currents and upwelling forecasting with the inclusion of the Stokes-Coriolis drift by a wave model, which induce an extra term of advection in the direction of wave group speed (Alari et al., 2016; Bruciaferri et al., 2021). Coupling also benefits wave modelling, for example where tidal currents modulate wave and wind activity (Renault et al., 2022; Valiente et al., 2021). Coupling an ocean with waves can have considerable impacts on SSTs, which can go in either direction, depending on the difference in momentum stress passed to the ocean (more momentum input by the waves in the case of Lewis et al. (2019), resulting in a near-surface cooling, but less momentum in Alari et al. (2016), resulting in warming), through modulation of the ocean stratification. Coupling a wave model with an atmospheric model will tend to decrease wind speed over young seas and increase ocean momentum flux, especially important during storms (Gentile et al., 2022; Bouin and Lebeaupin Brossier, 2020). In general, coupling will tend to dampen air-sea fluxes because components will tend to adjust to one another, so this may decrease ocean spread at the start of ensemble forecasts (Lea et al., 2022). However, the spread in SST will increase rapidly in regions which have a shallow surface mixed layer, which respond quickly to atmospheric spread (Lea et al., 2022). Precipitation and river flows can also have a local influence on near-surface temperatures and salinity in the ocean, especially during extreme precipitation events (Bouin et al., 202; Sauvage et al., 2018). The ocean can finally act as memory between two intense atmospheric events (e.g strong winds, and strong precipitation (Berthou et al., 2016; Lebeaupin Brossier et al., 2012) or in the case of marine heatwaves and extreme temperature or precipitation event (Berthou et al., 2024; Martín et al., 2024), in which cases a coupled system is beneficial for longer range forecasting. In regional atmospheric forecasts, using a predicted SST (either obtained through coupling or forcing) is beneficial for variables such as near-surface temperature (Mahmood et al., 2021), fog (Fallmann et al., 2019) or snow (Yamamoto et al., 2011).

However, it is worth noting that differences in near-surface parameterisations can also generate differences which are as large or larger than coupling differences (Gentile et al., 2022), indicating the need for continuous research and investment in observation systems of near-surface characteristics. Coupling is most successful when the water, heat and momentum budgets are closed, which can be challenging when model parameterisations are designed in forced mode. Recent parameterisation improvements taking into account coupled variables include wave coupling in the NEMO Turbulent Kinetic Energy scheme

(Couvelard et al., 2020), or current feedback taken into account in atmospheric turbulence (Renault et al., 2019), or finally the new Wave-Age dependent Stress Parameterisation (Bouin et al., 2024)). In some situations, increasing the complexity of air-sea exchanges can be beneficial, for example including sea spray effects on moisture and heat fluxes (Yang et al., 2019; Xu et al., 2021; Zhang et al., 2005; Bianco et al., 2011).

Coupling with land and river models are also attractive to provide river-flow forecasts, especially as the coupling interface gets more complex, to include back-water effects into rivers and coastal wetting and drying (Bianco et al., 2011). Finally, coupling with biogeochemistry and sediment transport models can provide interesting feedback on the ocean colour, with a feedback loop between thermal stratification and phytoplankton bloom, through the modulation of depth penetration of the solar heat flux (Skákala et al., 2022). Other feedbacks include chemistry and aerosols, where the atmosphere can then provide deposition fluxes (e.g. iron, nitrogen) to the ocean, and the phytoplankton sends back chemicals which can affect low-level cloud cover (Mulcahy et al., 2023).

The potential benefits of using a coupled framework are also reinforced by the move towards a multi-hazard approach to predictions. Natural hazards from multiple sources may combine or occur concurrently. Large waves, storm surges, high-wind speeds, and extreme precipitation are all hazards that are likely to co-occur, and influence each other through coupled feedback that can compound one another (for example through over-topping). Coupled systems that predict this feedback may enable an improvement in the range and consistency of actionable information to be provided through hazard warnings and guidance.

**3 How extended is the use of coupled modelling for Ocean Prediction?**

Many centres and research groups have developed monitoring and prediction tools independently for individual Earth components (e.g. atmosphere, ocean, land, waves, etc.). This is natural based on the historical context of their development and limitations on computing capabilities, but it has created an infrastructure within and across institutions that adds complexity to the task of unifying prediction systems. The major prediction centres are making progress towards an integrated approach by unifying software infrastructure for models and data assimilation capabilities as well as providing opportunities to increase interactions among the development teams of each system component. At the global scale, the use of atmosphere-ocean-sea-ice coupled model has increased rapidly in the past few years, usually starting with deterministic and then ensemble coupled capability: Allard et al., 2012, Komaromi et al., 2021 (Naval Research Laboratory), Mogensen et al. 2017 (European Centre of Medium Range Forecasting), Smith et al., 2018, Peterson et al., 2022 (Environment and Climate Change Canada), Guiavarc'h et al., 2019 (Met Office). In parallel, the perspective of seamless predictive capability (Ruti et al, 2020), especially important during impactful extreme cyclonic or convective events, means km-scale regional coupled systems are either operational (Durnford et al (2018) for the Great Lakes and Saint Lawrence river, Komaromi et al. 2021 for tropical cyclone regions) or are actively being developed in several centres or research group. Examples include in western Europe (Sauvage et al., 2021), southwestern Indian ocean (Corale et al., 2023) (the Northwest European shelf (Lewis et al., 2019); the northern Indian ocean (Castillo et al., 2022), the Red Sea (Rui et al.,2019, 2024). Finally, coupled river-ocean models, including two-

way coupling between river and ocean are used for operational forecasting of compound flooding under hurricanes in the Gulf of Mexico (Bao et al., 2024, using the COWAST system (Warner et al, 2010)).

The extent to the uptake of coupled modelling is still limited, however, by several barriers. First, it places extreme demands on computational resources: the cost of running an extra model is often prohibitive for agencies with limited forecasting remits (e.g. only ocean forecasting). However, recognising the benefits acknowledged above, these agencies are exploring alternatives, such as coupling with a single-column mixed layer model, either in the atmosphere or the ocean (Voldoire et al., 2017; Lemarié et al., 2021). For the agencies with several remits (e.g. weather, marine, hydrology, air quality forecasting), coupled modelling is more attractive and has the potential to reduce the complexity of the modelling chains, as well as prevent large data transfers between platforms.

A second major barrier is data assimilation, which requires the processing of environmental observations, is itself a technically challenging problem which is made harder if you try and harmonise that across all the earth system components. Ddata assimilation requires the calculation of an innovation (difference between the modelled and observed value) and then appropriately adjusting the model parameter space to create a state estimate that is optimised to best reflect understanding of model and observation errors. In coupled systems there are correlations between parameters in the different systems that need to be respected: for example, sea surface and air surface temperature are closely correlated. This creates an additional scientific and technical challenge that needs to be addressed in coupled forecasting systems (Penny and Hamill, 2017). The differing time scales inherent in ocean forecasting and atmospheric NWP is also problematic, though Lea et al. (2015) suggest using the shorter NWP based windows does allow for the retention of the longer oceanic time scales, as long as the memory inherited with cycling the system in time remains intact. Nevertheless, strongly coupled data assimilation means an observation in one model can be beneficial for both models (Fu et al. (2021), Phillipson et al, (2021)) and allows for coupled observation operators. Indeed, remote sensed observations of the ocean (remote sensed SST, radiances, colour, ice freeboard) require filtering out an atmospheric signal, a task which could be dealt with by a coupled assimilation system instead of externally, which potentially introduces contradictory biases from other systems.

Weaker barriers include the need for different frequency of running forecasts: ocean forecasts often run daily with a single deterministic member but the atmospheric and the wave forecasts require sub-daily ensembles with several members. In ensemble modelling, inflated spread schemes are often employed (e.g. in the SST), to generate a much larger spread than the ocean uncertainty, and must be modified in coupled systems (Lea et al., 2022). Nevertheless, the ocean and sea ice uncertainty needs thorough quantification against independent observational datasets for these schemes to be effective. Finally, simple bureaucratic barriers such as the constraint of a common forcing model in international projects can also prevent the adoption of coupled modelling.

## 4 Conclusion

Coupling models of different earth system components is a technical task which requires scientific software engineering expertise and high computing resources. Whilst common for seasonal and climate prediction, a handful of operational centres have achieved this for NWP timescales, most of them in the past five years. Coupling enables better treatment of air-sea interactions, especially important in the tropics, for intense events (tropical cyclones), for regions of strong SST gradients, eddies, tidal influence or complex coastlines. The cost is affordable for centres which have the responsibility for forecasting across different earth system components. In these cases, in addition to the benefits of coupled feedback, coupled forecasting allows forecast consistency, essential for impact-based forecasting of multi-hazard events. For other centres, cheaper solutions exist, such as only treating the boundary layer of the other earth system component, which is the most important part to coupled at short timescales.

Coupling models also increases knowledge exchange between researchers in different earth system components, which helps build our understanding of the earth system as a whole. Novel methods such as machine learning and artificial intelligence offer great hope in overcoming some of the barriers faced by traditional NWP. At a time of greater coupling between traditional numerical forecasting system, the use of ML/AI should cut across earth system components, and avoid the pitfalls of parameterisations designed with a single component in mind. This can only be achieved by a strong and organised coupling research community.

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

## Competing interests

The contact author has declared that none of the authors has any competing interests.

## Data and/or code availability

Not applicable.

## Authors contribution

JS started a draft of this document, SB took over and completed the article, helped with a literature review completed by VFL. PYLT, IH reviewed the text.

## Acknowledgements

The authors would like to acknowledge an anonymous reviewer and K. A. Peterson for their suggestions of improvement of the manuscript.