# Peer review of "Towards Earth System Modeling: Coupled Ocean Forecasting"

_State of the Planet, 2024_

## Referee Comment (RC2)

**Review of: Towards Earth System Modeling: Coupled Ocean Forecasting**

by Ségolène Berthou, John Siddorn, Vivian Fraser-Leonhardt, Pierre-Yves Le Traon, Ibrahim Hoteit

October 22, 2024

**Manuscript Synopsis**

The article is a review of the ongoing trend toward the use of coupled prediction in ocean forecasting. As I am to understand, this is part of a series of reviews in a guide to the operational oceanography value chain. The article is a wonderful view of the need for coupled ocean prediction, the potential and potential benefit of integrating ocean prediction with existing atmospheric and hydrological prediction value chains and infrastructure, and finally some of the challanges to coupled prediction, and in particular coupled prediction including coupled assimilation. I would only *suggest* a few minor revisions to the article. My sole complaint would be while the cited Brassington et al. [2015] laid out the aspirations of the ocean prediction community to embrace coupled forecasting almost a decade ago, this article perhaps does not give enough credit to the various operational centres and systems that have managed to make progress on this front in the intervening period – my *Minor Comment* 2

**My recommendation is Minor Revisions**

**Major Comments**

None.

**Minor Comments**

1. I will begin at the beginning with the abstract's opening words: "The work we do is hard." Okay, I have paraphrased that somewhat for effect. That phrase, for me, evoked an image of a cartoon by Nathan W. Pyle posted on our coffee room bulletin board that "Science is difficult." (`https://www.facebook.com/nathanwpyle2/posts/466709991490794/` ; I apologize if the link is broken, but neither do I want the Journal subject to copyright violations.) More particularly, it eludes to a sediment that what we do is hard – and we should not particularly expect success, or be disappointed in a lack of success. A manuscript's abstract tends to be fairly personal, and I am not going to suggest the authors change this, as I imagine a lot of thought went into starting the abstract in this fashion. I just thought it good to remind that words can sometimes be read in unintended ways, with unintended consequences. Personally, I might have gone down a route that great strides have been made in ocean forecasting, but future advancement of our work, and our earth system prediction colleagues work, will require a coupled approach.

2. Brassington et al. [2015] lay out some of the intentions of the community a somewhat dated decade ago. Although the use of coupled seasonal and climate predictions was then, and is more so now, fairly ubiqious. Coupled short range and Numerical Weather Prediction (NWP) predictions are still a relative rarity, but they do exist, and probably should be given some credence, [e.g. Komaromi et al., 2021, Mogensen et al., 2017, Smith et al., 2018, Peterson et al., 2022], but I am sure the authors' literature review can identify some more (even if one has to resort to technical reports).

3. SST (Sea Surface Temperature) is not defined before its first use.

4. ll. 126-129. It is perhaps worth mentioning the ECMWF approach of integration into a single executable might be detrimental to open source / code sharing requirements, even if just one of the components is propriety code.

5. The authors discuss barriers to coupled data assimilation, particularly with regards to the added complications of cross model covariances in strongly coupled data assimilation. Unmentioned are other barriers, such as the differing time scales inherent in ocean forecasting and atmospheric NWP – likely further exasperated with the inclusion of land surface/hydrological modelling and biogeochemistry. However, Lea et al. [2015] does suggest using the shorter NWP based windows does allow for the retention of the longer oceanic time scales, as long as the memory inherit with cycling the system in time remains intact.

6. Futhermore, the authors do not mention some potential advantanges of coupled data assimilation, beyond the obvious acheivement of a more balanced initial state: Coupled data assimilation allows for coupled observation operators. Data assimilation of remote sensed SST, and more particularly remote sensed radiances, is inherently a coupled problem with the observed radiance a function of the SST and the atmospheric transmission, existing strategies (i.e. using processed SST retrievals) leave open the possibility of introducing external, and potentially contradictory biases from other systems. Similar advantages also exist with ice freeboard measurements (dependent on ice thickness and snow thickness), or even for remote sensing of ocean colour (dependent on ocean colour and atmospheric moisture; personal communications). Further examples likely exist outside my realm of knowledge. Again, this would be an advantage, allowing a fully self-consistent observation, with potential for a better and more self-consistent estimated state, although hardly a trivial exercise.

7. Spread and initial condition uncertainty (ll. 146-149). While I would agree atmospheric spread inflation schemes can often inflate ocean spread (SST) beyond initial condition uncertainty, I would also argue that quite often ocean spread does not adequately represent observed uncertainty. Peterson et al. [2022] showed that sea ice initial conditions failed to adequately represent the uncertainty in the estimation of the sea ice state. While that was in the case of a deterministic ocean and sea ice initial state used for ensemble forecasting, similar underestimation of the uncertainty exists in ensemble initializations: Sea ice perturbations in Zuo et al. [2017] are achieved by randomly sampling high resolution OSTIA sea ice concentrations into the lower resolution ORAS5 ensemble, however Renfrew et al. [2021] suggest the sea ice edge in OSTIA is too wide, owing to the large footprint of the SSMIS retrievals of the OSTIA assimilated OSISAF sea ice analysis. Randomly sampling a high resolution product, whose effective resolution is much coarser, is not going to adequately sample the uncertainty in sea ice concentration. Without any definite example, I would suggest at least for instances when an SST analysis is assimilated (which excludes [Lea et al., 2022]), similar reliance on a single smoothed analysis might lead to an under-representation of the SST observation uncertainty in an initial spread of SST – although here, the inherent smoothing of the SST analysis is not as obvious – the microwave satellite footprints are actually quite high resolution – it is the correction of (due to atmospheric transmission) bias, anchored by more sparse insitu measurements, that likely leads to the smoothing of the analysis.

8. Please do not forget to fill in (or remove) the acknowledgements section.

**References**

G.B. Brassington, M.J. Martin, H.L. Tolman, M. Balmeseda S. Akella, C.R.S. Chambers, E. Chassignet, J.A. Cummings, Y. Drillet, P.A.E.M. Jansen, P. Laloyaux, D. Lea, A. Mehra, I. Mirouze, H. Ritchie, G. Samson, P.A. Sandery, G.C. Smith, M. Suarez, and R. Todling. Progress and challenges in short- to medium-range coupled prediction. *Journal of Operational Oceanography*, 8(sup2):s239–s258, 2015. doi: 10.1080/1755876X.2015.1049875. URL https://doi.org/10.1080/1755876X.2015.1049875.

William A. Komaromi, Patrick A. Reinecke, James D. Doyle, and Jonathan R. Moskaitis. The naval research laboratory's coupled ocean-atmosphere mesoscale prediction system-tropical cyclone ensemble (coamps-tc ensemble). *Weather and Forecasting*, 36(2):499 – 517, 2021. doi: 10.1175/WAF-D-20-0038.1. URL `https://journals.ametsoc.org/view/journals/wefo/36/2/WAF-D-20-0038.1.xml`.

D. J. Lea, I. Mirouze, M. J. Martin, R. R. King, A. Hines, D. Walters, and M. Thurlow. Assessing a new coupled data assimilation system based on the met office coupled atmosphere?land?ocean?sea ice model. *Monthly Weather Review*, 143(11):4678 – 4694, 2015. doi: 10.1175/MWR-D-15-0174.1. URL `https://journals.ametsoc.org/view/journals/mwre/143/11/mwr-d-15-0174.1.xml`.

Daniel J. Lea, James While, Matthew J. Martin, Anthony Weaver, Andrea Storto, and Marcin Chrust. A new global ocean ensemble system at the met office: Assessing the impact of hybrid data assimilation and inflation settings. *Quarterly Journal of the Royal Meteorological Society*, 148(745):1996–2030, 2022. doi: https://doi.org/10.1002/qj.4292. URL `https://rmets.onlinelibrary.wiley.com/doi/abs/10.1002/qj.4292`.

Kristian S. Mogensen, Linus Magnusson, and Jean-Raymond Bidlot. Tropical cyclone sensitivity to ocean coupling in the ecmwf coupled model. *Journal of Geophysical Research: Oceans*, 122(5):4392–4412, 2017. doi: https://doi.org/10.1002/2017JC012753. URL `https://agupubs.onlinelibrary.wiley.com/doi/abs/10.1002/2017JC012753`.

K. Andrew Peterson, Gregory C. Smith, Jean-François Lemieux, François Roy, Mark Buehner, Alain Caya, Pieter L. Houtekamer, Hai Lin, Ryan Muncaster, Xingxiu Deng, Frédéric Dupont, Normand Gagnon, Yukie Hata, Yosvany Martinez, Juan Sebastian Fontecilla, and Dorina Surcel-Colan. Understanding sources of northern hemisphere uncertainty and forecast error in a medium-range coupled ensemble sea-ice prediction system. *Quarterly Journal of the Royal Meteorological Society*, 148(747):2877–2902, 2022. doi: https://doi.org/10.1002/qj.4340. URL `https://rmets.onlinelibrary.wiley.com/doi/abs/10.1002/qj.4340`.

I. A. Renfrew, C. Barrell, A. D. Elvidge, J. K. Brooke, C. Duscha, J. C. King, J. Kristiansen, T. Lachlan Cope, G. W. K. Moore, R. S. Pickart, J. Reuder, I. Sandu, D. Sergeev, A. Terpstra, K. Våge, and A. Weiss. An evaluation of surface meteorology and fluxes over the iceland and greenland seas in ERA5 reanalysis: The impact of sea ice distribution. *Quarterly Journal of the Royal Meteorological Society*, 147(734):691–712, 2021. doi: https://doi.org/10.1002/qj.3941. URL `https://rmets.onlinelibrary.wiley.com/doi/abs/10.1002/qj.3941`.

Gregory C. Smith, Jean-Marc Bélanger, François Roy, Pierre Pellerin, Hal Ritchie, Kristjan Onu, Michel Roch, Ayrton Zadra, Dorina Surcel Colan, Barbara Winter, Juan-Sebastian Fontecilla, and Daniel Deacu. Impact of coupling with an ice–ocean model on global medium-range nwp forecast skill. *Monthly Weather Review*, 146(4):1157–1180, 2018. doi: 10.1175/MWR-D-17-0157.1. URL `https://doi.org/10.1175/MWR-D-17-0157.1`.

Hao Zuo, Magdalena Alonso-Balmaseda, Eric de Boisseson, S Hirahara, Marcin Chrust, and Patricia de Rosnay. A generic ensemble generation scheme for data assimilation and ocean analysis. Technical report, ECMWF, 2017. URL `https://www.ecmwf.int/node/17831`.

---

## Author Response (AR1)

We thank both reviewers for their interest in our work and their useful comments. It made us aware of very interesting points and made us look more in depth into the "coupling" literature. We believe the manuscript is now improved. Please see our replies in this colour.

**Reviewer #1:**

The brief review paper is quite well written.

However, it misses a conclusion section combining the information in section 2 (why coupling is important) and section 3 (who is doing coupling and to what degree) in a section containing recommendations on what is be the road forward for the various ocean forecasting centers. This section could also highlight any additional research needed if we do not have the information needed to make these recommendations.

This point should be suggested as a suggestion to improve the manuscript. It is up to the authors to decide if they want to follow it.

We thank the reviewer for their time and recommendation. We added a conclusion section combining sections 2 and 3, it is a good addition to the manuscript, thank you.

**Reviewer#2**

**Manuscript Synopsis**

The article is a review of the ongoing trend toward the use of coupled prediction in ocean forecasting. As I am to understand, this is part of a series of reviews in a guide to the operational oceanography value chain. The article is a wonderful view of the need for coupled ocean prediction, the potential and potential benefit of integrating ocean prediction with existing atmospheric and hydrological prediction value chains and infrastructure, and finally some of the challanges to coupled prediction, and in particular coupled prediction including coupled assimilation. I would only suggest a few minor revisions to the article. My sole complaint would be while the cited Brassington et al. [2015] laid out the aspirations of the ocean prediction community to embrace coupled forecasting almost a decade ago, this article perhaps does not give enough credit to the various operational centres and systems that have managed to make progress on this front in the intervening period – my Minor Comment 2.

My recommendation is Minor Revisions

**Major Comments**

None.

**Minor Comments**

1. I will begin at the beginning with the abstract's opening words: "The work we do is hard." Okay, I have paraphrased that somewhat for effect. That phrase, for me, evoked an image of a cartoon by Nathan W. Pyle posted on our coffee room bulletin board that "Science is difficult." (https://www.facebook.com/nathanwpyle2/posts/466709991490794/ ; I apologize if the link is broken, but neither do I want the Journal subject to copyright violations.) More particularly, it eludes to a sediment that what we do is hard – and we should not particularly expect success, or be disappointed in a lack of success. A manuscript's abstract tends to be fairly personal, and I am not going to suggest the authors change this, as I imagine a lot of thought went into starting the abstract in this fashion. I just thought it good to remind that words can sometimes be read in unintended ways, with unintended consequences. Personally, I might have gone down a route that great strides have been made in ocean forecasting, but future advancement of our work, and our earth system prediction colleagues work, will require a coupled approach.

**Thank you, it's a very good point, we've changed the first sentence of the abstract, which is now much more positive.**

2. Brassington et al. [2015] lay out some of the intentions of the community a somewhat dated decade ago. Although the use of coupled seasonal and climate predictions was then, and is more so now, fairly ubiqious. Coupled short range and Numerical Weather Prediction (NWP) predictions are still a relative rarity, but they do exist, and probably should be given some credence, [e.g. Komaromi et al., 2021, Mogensen et al., 2017, Smith et al., 2018, Peterson et al., 2022], but I am sure the authors' literature review can identify some more (even if one has to resort to technical reports).

**Thank you, we largely extended the references in this section (lines 135-150 in the tracked changed document).**

3. SST (Sea Surface Temperature) is not defined before its first use.

**Thank you.**

4. ll. 126-129. It is perhaps worth mentioning the ECMWF approach of integration into a single executable might be detrimental to open source / code sharing requirements, even if just one of the components is propriety code.

**Thank you, we did not approach the code propriety aspect, so we left this comment out.**

5. The authors discuss barriers to coupled data assimilation, particularly with regards to the added complications of cross model covariances in strongly coupled data assimilation. Unmentioned are other barriers, such as the differing time scales inherent in ocean forecasting and atmospheric NWP – likely further exasperated with the inclusion of land surface/hydrological modelling and biogeochemistry. However, Lea et al. [2015] does suggest using the shorter NWP based windows does allow for the retention of the longer oceanic time scales, as long as the memory inherit with cycling the system in time remains intact.

**Thank you, we added a sentence on this, lines 169-172.**

6. Futhermore, the authors do not mention some potential advantanges of coupled data assimilation, beyond the obvious acheivement of a more balanced initial state: Coupled data assimilation allows for coupled observation operators. Data assimilation of remote sensed SST, and more particularly remote sensed radiances, is inherently a coupled problem with the observed radiance a function of the SST and the atmospheric transmission, existing strategies (i.e. using processed SST retrievals) leave open the possibility of introducing external, and potentially contradictory biases from other systems.

Similar advantages also exist with ice freeboard measurements (dependent on ice thickness and snow thickness), or even for remote sensing of ocean colour (dependent on ocean colour and atmospheric moisture; personal communications). Further examples likely exist outside my realm of knowledge. Again, this would be an advantage, allowing a fully self-consistent observation, with potential for a better and more self-consistent estimated state, although hardly a trivial exercise.

**Thank you, this is very interesting, we were not aware of this advantage, but it makes a lot of sense. We have also added a sentence here: lines 172-176.**

7. Spread and initial condition uncertainty (ll. 146-149). While I would agree atmospheric spread inflation schemes can often inflate ocean spread (SST) beyond initial condition uncertainty, I would also argue that quite often ocean spread does not adequately represent observed uncertainty. Peterson et al. [2022] showed that sea ice initial conditions failed to adequately represent the uncertainty in the estimation of the sea ice state. While that was in the case of a deterministic ocean and sea ice initial state used for ensemble forecasting, similar underestimation of the uncertainty exists in ensemble initializations: Sea ice perturbations in Zuo et al. [2017] are achieved by randomly sampling high resolution OSTIA sea ice concentrations into the lower resolution ORAS5 ensemble, however Renfrew et al. [2021] suggest the sea ice edge in OSTIA is too wide, owing to the large footprint of the SSMIS retrievals of the OSTIA assimilated OSISAF sea ice analysis. Randomly sampling a high resolution product, whose effective resolution is much coarser, is not going to adequately sample the uncertainty in sea ice concentration.

Without any definite example, I would suggest at least for instances when an SST analysis is assimilated (which excludes [Lea et al., 2022]), similar reliance on a single smoothed analysis might lead to an under-representation of the SST observation uncertainty in an initial spread of SST – although here, the inherent smoothing of the SST analysis is not as obvious – the microwave satellite footprints are actually quite high resolution – it is the correction of (due to atmospheric transmission) bias, anchored by more sparse insitu measurements, that likely leads to the smoothing of the analysis.

**Thank you again for this point, this is a very interesting point, very useful to be aware of it. We added a line, l183.**

8. Please do not forget to fill in (or remove) the acknowledgements section.

We have done so.